# L-CPPA: Lattice-based conditional privacy-preserving authentication scheme for fog computing with 5G-enabled vehicular system

**Abdulwahab Ali Almazroi**[1]*, **Mohammed A. Alqarni**[2], **Mahmood A. Al-Shareeda**[3]*, **Selvakumar Manickam**[4]

**1** Department of Information Technology, College of Computing and Information Technology at Khulais, University of Jeddah, Jeddah, Saudi Arabia, **2** Department of Software Engineering, College of Computer Science and Engineering, University of Jeddah, Jeddah, Saudi Arabia, **3** Department of Communication Engineering, Iraq University College (IUC), Basra, Iraq, **4** National Advanced IPv6 Centre (NAv6), Universiti Sains Malaysia, USM, Penang, Malaysia

☯ These authors contributed equally to this work.
* aalmazroi@uj.edu.sa (AAA); alshareeda022@gmail.com (MAAS)

**Data Availability Statement:** All relevant data are within the paper.

## Abstract

The role that vehicular fog computing based on the Fifth Generation (5G) can play in improving traffic management and motorist safety is growing quickly. The use of wireless technology within a vehicle raises issues of confidentiality and safety. Such concerns are optimal targets for conditional privacy-preserving authentication (CPPA) methods. However, current CPPA-based systems face a challenge when subjected to attacks from quantum computers. Because of the need for security and anti-piracy features in fog computing when using a 5G-enabled vehicle system, the L-CPPA scheme is proposed in this article. Using a fog server, secret keys are generated and transmitted to each registered car via a 5G-Base Station (5G-BS) in the proposed L-CPPA system. In the proposed L-CPPA method, the trusted authority, rather than the vehicle's Onboard Unit (OBU), stores the vehicle's master secret data to each fog server. Finally, the computation cost of the suggested L-CPPA system regards message signing, single verification and batch verification is 694.161 ms, 60.118 ms, and 1348.218 ms, respectively. Meanwhile, the communication cost is 7757 bytes.

## 1 Introduction

After the World Health Organization indicated an increase in the number of deaths and accidents on the roads, intelligent transportation systems (ITS) became the interest of academic scholars and companies [1–3]. The Fifth-Generation (5G)-assisted vehicular system with supporting computing based fog is the most promising in the domain of ITS in order to preserve people, regulate the road environment and manage traffic jams. The number of cars in cities is rising rapidly, and drivers' needs are becoming more varied [4–6].

Consequently, the 5G mobile networks supporting vehicular communication can provide the necessary high bandwidth and extensive coverage in the present day's applications. It offers a variety of difficulties and potential benefits for road networks. For 5G wireless networks, the

**Funding:** The authors extend their appreciation to the Deputyship for Research \& Innovation, Ministry of Education in Saudi Arabia for funding this research work through project number MoE-IF-UJ-22-04100409-7.

**Competing interests:** The authors have declared that no competing interests exist.

maximum data transfer rate is 20 Gb/s, that the medium value is 100 Mb/s [7, 8]. The essential goal of fog-based computing is not only to boost the system's processing capability but also to lower the system's return pressure and improve the user's service experience by processing data locally at the vehicle terminal rather than sending it to the trusted authority in the network's distant core [9–11].

Information overflow due to redundant data may occur in autonomous vehicles because of the necessity for real-time sensing, calculation, and communication. In addition, the limited range of current communication technologies makes it challenging to anticipate traffic conditions outside of a vehicle's line of sight. Autonomous driving conditions have seen the development of digital twin systems to address these concerns [12]. Sensitive information about drivers is easily intercepted and altered since the data created by automobiles in motion with servers is carried on public networks. Furthermore, servers are put under extreme strain by the vast quantities of true-time data produced by these cars, equipment, users, passengers, and other social links. As a result, Chen et al. [13] presented a secure authentication technique that makes use of private cloud infrastructures. In addition, they provided a more efficient key transport stage to lessen the burden of computation on servers of cloud.

To cause havoc on the roads, hackers can alter, counterfeit, replay, or otherwise tamper with the messages vehicles communicate with one another via computing-based fog with 5G-assisted vehicular network. Information about the car (its location, the quality of the roads, etc.) is included in these transmissions [14, 15]. As a result, these communications need to be protected and taken against the third party before the transportation system can be put into action.

Many conditional privacy-preserving authentication (CPPA) techniques, however, have been presented as ways to encrypt and safeguard communications between cars. Methods based on Certificateless-Based, Identity (ID) (Chebyshev Polynomial, Elliptic Curve Cryptography (ECC), and Bilinear Pair Cryptography (BPC)), Group Signature (GS), and Public Key Infrastructure (PKI) are frequently employed in CPPA. Most current systems are constructed with primitives based on standard cryptography, such as Elliptic Curve Diffie-Hellman or Diffie-Hellman, making them vulnerable to quantum attacks.

Consequently, the intent of this research is to propose a lattice-based CPPA method for 5G-assisted vehicular system with fog computing in order to counteract quantum attacks utilising post-quantum cryptography techniques. The following are the major results of our solution work.

- In this paper, we take a look back and examine a sophisticated taxonomy of the existing authentication and conditional privacy-preserving schemes based on the approaches used: Certificateless-Based, ID (Chebyshev Polynomial, ECC and BPC), GS, and PKI.

- This research offers a system model for 5G-assisted vehicular system with fog computing by substituting a fog server for cloud computing throughout the authentication process and the coverage zone of 5G-BS for the limited reach of Road-Side Units (RSUs).

- For 5G-assisted vehicle fog computing, this work offers a lattice-based conditional privacy-preserving authentication (L-CPPA) approach to succeed anti-piracy and security features.

- In this research, we analyse the security of the proposed L-CPPA system and show how efficient it is in terms of communicational and computational overheads.

Our remaining tasks are divided into several parts: Section 2 proposes the classification of the related works. Section 3 provides the background of the vehicular network. The solution's six algorithms are proposed in Section 4. The analysis of security and efficient of performance

of this paper are provided in Section 5 and 6, respectively. Section 7 shows the conclusion of this work.

## 2 Related work

Some existing authentication and privacy-preserving systems suggested in order to secure vehicular networks. As presented in Fig 1, this paper provides a sophisticated taxonomy of systems as follows.

### 2.1 Public Key Infrastructure (PKI)

Several researchers [16–26] have proposed PKI systems to provide authentication and privacy-preserving attributes for the vehicular network. During the early 2000s, Raya et al. [16] and Hubaux et al. [17] investigated various privacy and security attributes and issues related to intelligent vehicular. Nevertheless, PKI-based schemes need a massive number of key pairs and the pertinent certificates of identity to be preloaded in each registered vehicle. The main reason for preloading a large number of parameters is to keep the node's identification. Due to a vehicle having to randomly select a key pair to preserve message authenticity and integrity attributes in every communication, the expense of storing vehicles and trusted authority (TA) is high. Meanwhile, it is difficult for the TA to determine a malevolent adversary's true identity.

### 2.2 Group Signature (GS)

As a concept, Group Signature (GS) was proposed for the first time by van Heyst and Chaum [27]. Joiner of a domain can sign documents on behalf of the whole without revealing their identities. Many academics [28–32] have developed GS-based methods to deal with the problems that can occur with PKI-based systems. However, due to the rise in the number of banned cars, such a strategy necessitates a massive expansion of the Certification Revocation List

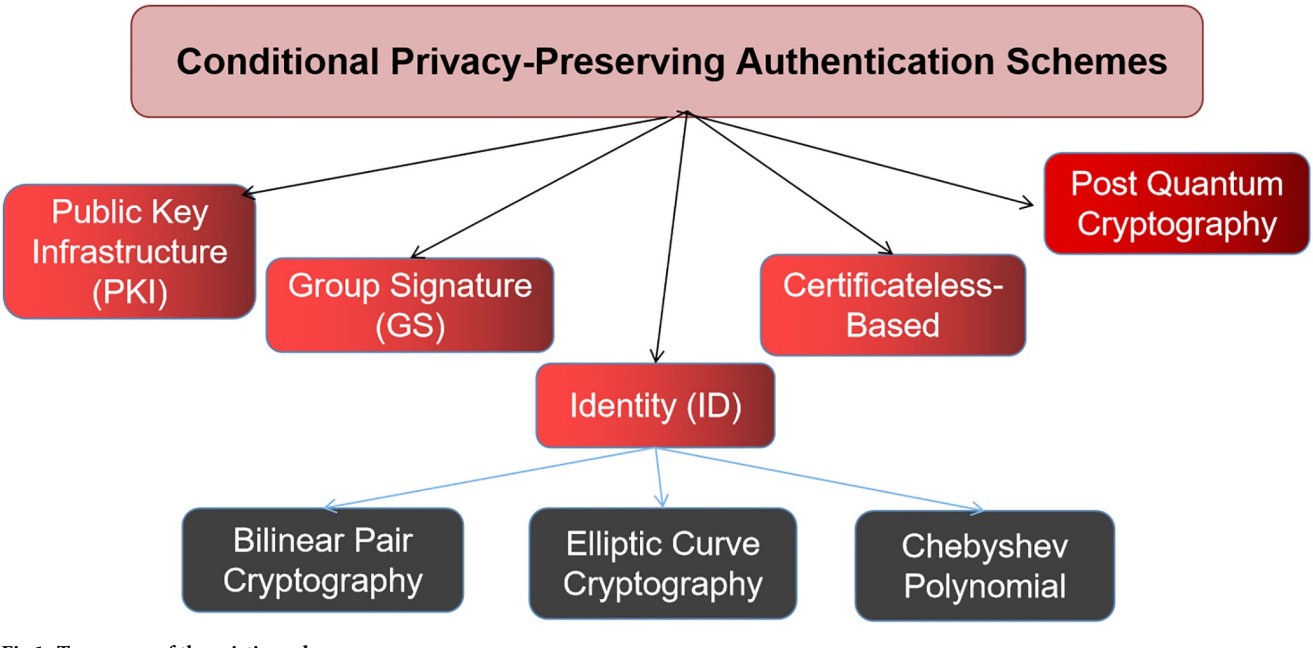

**Fig 1. Taxonomy of the existing schemes.**

(CRL). In addition, there are large communication and computation costs associated with the two pairing-based procedures.

## 2.3 Identity (ID)

Numerous scholars have proposed ID based systems to improve upon the drawbacks of the aforementioned (e.g., PKI and GS methods) and provide extensive protections for vehicular networks. For ID-based schemes, the public key is derived from the node's identifier and the master data is generated by a TA. Schemes are classified according to whether they use bilinear pair cryptography, elliptic curve cryptography, or the Chebyshev polynomial. These methods are discussed in this publication.

**2.3.1 Bilinear pair cryptography.** Several researchers [33–39] have used operations correlating with BPC to verify and sign signatures. BPC is effective in signing and verifying signatures, however, the processes involved are complex and time-consuming, resulting in significant performance efficiency losses. Message verification in the Pournaghi et al. [39] system requires three bilinear pair processes and one hashing on the side of the checker, while message signing only requires one bilinear pair procedure and one hashing of map-based point on the part of the contributing registered vehicle.

**2.3.2 Elliptic curve cryptography.** To avoid using time-consuming operations associated with bilinear pair and Point-to-Map hashing function, He et al. [40] constructed an ID-founded system applying lightweight operations associated with ECC for message signature verification. Additionally, several researchers [41–47] have proposed an authentication schemes based on ECC. Al-Shareeda et al. [45] use ECC without the road-side unit (RSU) to secure communications among vehicles in a 5G-enabled vehicular network. This scheme needs four scalar point multiplication operations to verify a single message shared among vehicles. The scheme of Al-Shareeda et al. [45] suffer from enormous overhead regards communicational and computational costs.

**2.3.3 Chebyshev polynomial.** Several scholars [31, 48, 49] have proposed employing operations based Chebyshev polynomial in terms of semi-group and chaotic to reduce the need for a high number of operations in elliptic curve cryptography. To protect data transmissions in 5G-assisted vehicular system with fog computing, Cui et al. suggested a Chebyshev polynomial-based approach [31]. In this setup, fog servers are used as a link in the chain between automobiles and TA.

## 2.4 Certificateless-based

The escrow problem is the ID-based solutions' primary drawback. Therefore, several researchers [50–52] have proposed a certificateless-based scheme to secure communication. Mei et al. [52] constructed the overall signature approach with conditional privacy preservation that is certificateless. This scheme utilises the complete aggregation approach to cut down on processing costs before achieving conditional privacy preservation through the usage of a pseudonym mechanism.

## 2.5 Post quantum cryptography

Nevertheless, the majority of existing schemes (including those discussed above) are built using conventional cryptographic primitives like Diffie-Hellman or Elliptic Curve Diffie-Hellman. Such schemes are well known to be vulnerable to quantum attacks. Thus, these schemes [53–55] are designed to mitigate quantum attacks. Utilizing lattice, Mukherjee et al. [53] developed a batch-verifiable authentication scheme for vehicular networks. Dharminder et al. [54] suggested an authentication and privacy system based on lattice that provides batch

verification of message revocation and multiple signatures. One drawback of such lattice-based schemes in [53, 54] is that each OBU stores the master data of the TA, opening up a potential attack vector. Li et al. [55] designed a lattice-based scheme to simultaneously provide mutual authentication and privacy-preserving. Li et al. [55] suffered to update the secret key of OBU's vehicle during online mode and are vulnerable to satisfying revocation attribute.

Based on our knowledge, this is the first proposed authentication system with privacy-preserving using lattice cryptography for fog computing with 5G-enabled VANET, which these previous studies are vulnerable to quantum attacks since using traditional cryptography. This work, therefore, offers a lattice-based conditional privacy-preserving authentication (L-CPPA) strategy for 5G-assisted vehicle system with fog computing. To overcome the restriction introduced by [55], the suggested L-CPPA system uses a fog server to produce and transmit a secret key for each enrolled car via 5G-BS. To overcome the restriction in [53, 54], the TA in the suggested L-CPPA system stores its master secret data to each fog server rather than the OBU of the vehicle.

## 3 Preliminaries

In this section, the proposed system model, mathematical used, and design goal are offered in detail as follows.

### 3.1 Proposed system model

The proposed system paradigm for vehicle networks is laid out in this part, and it makes use of fog servers and 5G technologies. In order to authenticate users, our suggested system uses a fog server instead of cloud computing [31]. To counteract the limited range of the Road-Side Unit (RSU) we propose using 5G-BS for our communications [48]. The four essential parts of our suggested system model are the 5G-Base Station (5G-BS), Trusted Authority (TA), On-Board Unit (OBU) and the fog server, as illustrated in Fig 2. The following is an explanation of how each part performs its designated task.

- Trusted Authority (TA): In this system model, the TA serves as a reliable third party and has strong computational capabilities. It is the responsibility of the TA to create system parameters and preload these data into each fog server and vehicle offline. As a result, the TA is the one who can determine each vehicle's true identity via a supporting fog server from the replying communication data. In this paper, we assume the TA component is very strong and reliable from masquerading attacks.

- 5G-Base Station (5G-BS): The 5G-BS is frequently used as a radio apparatus along the side of the road. In our proposed, 5G-BS doesn't do any storage and computation method.

- Fog Server: The fog server is frequently utilised as a wireless device along the 5G-BS. Normally, the fog server communicates with vehicles via the 5G protocol via 5G-BS. The fog server must validate messages sent by vehicles during communication. After that, it either processes those signals locally or transmits them to TA.

- On-Board Unit (OBU): The system model requires that every vehicle have an OBU. Its is a juggle-proof system that prevents data from ever leaking. Additionally, using the DSRC and 5G protocols, the OBU might offer wireless technology among vehicles or a close fog server.

### 3.2 Mathematical used

As in [56], uppercase strong letters represent matrices, while lowercase bold letters represent vectors. A set, such as $E$, is written in bold italic uppercase. Meanwhile, $\mathbb{R}$ and $\mathbb{Z}$

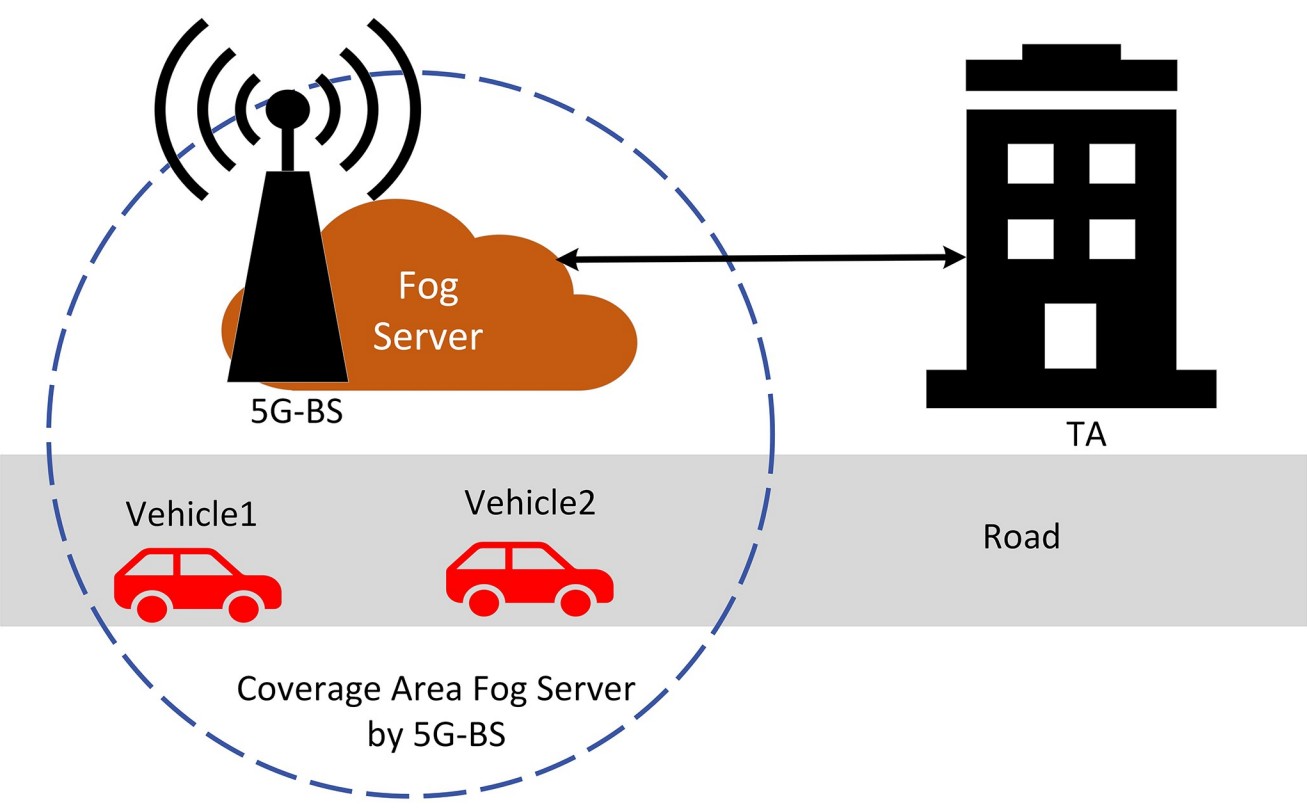

**Fig 2. Proposed system model.**

indicate the set of real numbers and the set of integers, receptively. [N] indicates the set $\{1, 2, . . ., N\}$. The following subsection, lattice and Gaussian distribution are provided in detail.

**3.2.1 Lattice.** The Euclidean space $\mathbb{R}^s$'s discrete additive subgroup is known as an $s$-dimensional lattice $\wedge$. It supposes that $\Upsilon_1, \Upsilon_2, .., \Upsilon_s$ are $s$ linearly distinct elements in $\mathbb{R}^s$. If any item in the lattice $\wedge$ can be calculated from $\Upsilon_1, \Upsilon_2, .., \Upsilon_s$, those linearly distinct elements $\Upsilon_1, \Upsilon_2, .., \Upsilon_s$ are a foundation of $\wedge$. Meanwhile, the foundation matrix of that lattice $\wedge$ is identified as $\mathbf{E} = (\Upsilon_1, \Upsilon_2, .., \Upsilon_s)$. Formally, $\wedge$ is a set identified as follows:

$$\wedge \mathbf{E} = \wedge(\Upsilon_1, \Upsilon_2, .., \Upsilon_s) = (\sum_{i=1}^{s} x_i \Upsilon_i, x_i \in \mathbb{Z}) \tag{1}$$

- Challenge of Small Integer Solution (SIS): Given a real number $\alpha$, a positive integer $q$, and a random matrix $\mathbf{A} \in \mathbb{Z}_q^{\beta*a}$, the problem of SIS is to resolve a non-zero element $\varsigma \in \mathbb{Z}^a$ for instance $\mathbf{D}\varsigma = 0 \ mod q$ with $\|\varsigma\| \leq \alpha$. Particularly, for an element $c \in \mathbb{Z}^s$ of non-zero.

- Challenge of Inhomogeneous Small Integer Solution (ISIS): The challenge of ISIS is to resolve a vector $\varsigma \in \mathbb{Z}^a$ for instance $\mathbf{D}\varsigma = c \ mod \ q$, that $\|\varsigma\| \leq \alpha$.

It should be noted that our system model for fog computing with a 5G-based vehicular system relies heavily on the SampleD and TrapGen algorithms. We shall demonstrate the pertinent lemmas regarding these approaches as follows:

- Lemma 1 [57]: Supposed a safeguarding element $s$ and two integers $q \geq 3$, $a > 5nlogq$, a random matrix $\mathbf{D} \in \mathbb{Z}_q^{a*\beta}$ can be generated with the help of the probabilistic polynomial time technique TrapGen and a foundation $\subset \wedge(\mathbf{D})$. Over, the inequality $T_D\| \leq O(slogq)$ holds with massive likelihood.

- Lemma 2 [58]: Supposed a safeguard element $e$, a main element $q$, a true element $\delta$, a value $a \geq \beta$ and duo matrices $\mathbf{D} \ in\mathbb{Z}_q^{a*\beta}$, $T_D \in \mathbb{Z}_q^{a*a}$. A lattice $\wedge(\mathbf{D})$ is defined by the matrix $\mathbf{D}$. $T_D$ is the foundation of this cryptography. if $\delta \geq \|T_D\|w(\sqrt{logs})$, for any element $\varsigma \in \mathbb{Z}_q^n$, methods based on SampleD probabilistic polynomials can produce an element at a certain time $v \in \mathbb{Z}_q^a$ such that $\mathbf{D}$ v $= \varsigma$ mod q. Meantime, the inequality $\|v\| \leq \delta\sqrt{a}$ ok with negligibleless likelihood.

**3.2.2 Distribution of Gaussian.**   The Gaussian distribution on $\mathbb{R}^a$ is identified by the following formula for a item $c \in \mathbb{R}^a$ and a ideal value $\delta > 0$

$$\vee x \in \mathbb{R}^a, \alpha\delta, s(X) = (\frac{1}{\sqrt{2\pi\delta^2}})^\beta exp(\frac{-\|w - s\|^2}{2\delta^2}) \tag{2}$$

$\delta$ is the Gaussian distribution's level deviation, while c is its mean element.

- Discrete Gaussian Distribution: The Gaussian distribution on a lattice $\wedge$ is denoted by the expression $\alpha\delta$, $s(\wedge) = \Sigma_{w\in\wedge}(w)$. In order to calculate the lattice $\wedge$'s discrete Gaussian distribution, we use the following formulas for the mean number $t$ and the level deviation $\delta$:

$$\mathbb{D}_{\wedge,\delta,t} = \frac{\wedge, \delta, t(w)}{\wedge, \delta, s(\wedge)} \tag{3}$$

As an example, when $c = 0$, the distribution of discrete Gaussian is expressed as $\mathbb{D}_{\wedge,\delta,t}$. The following are two guiding principles that completely describe the distribution of discrete Gaussian.

- Lemma 3 [59]: The standard deviation is a positive real value $\delta$. It is possible to draw the following conclusion for any positive value $k > 1$.

$$Pr_{\lambda\leftarrow\mathbb{D}_{\wedge,\delta}}[\|\lambda\| > k.\delta] \leq 2e^{\frac{-k^2}{2}} \tag{4}$$

- Lemma 4 [59]: For any element $v \in \mathbb{Z}_\beta$, the distribution of Gaussian item $\delta$ is equal to $\delta = w(\|v\|\sqrt{lognlog(N/(N - 1))})$, provided that $l$, $s$ and $N$ are all positive integers.

$$Pr_{z\leftarrow\mathbb{D}_{\wedge,\delta}}[\frac{\mathbb{D}_{\wedge,\delta}(\lambda)}{\mathbb{D}_{\wedge,\delta}v(\lambda)} = \theta(\frac{N}{(N - 1)})] = 1 - l.2^{-w(logn)} \tag{5}$$

- Rejection Sampling Lemma [59]: It supposes where $V$ is a class of $\mathbb{Z}_\beta$, in which the norm of any item is less than $T$. Meanwhile, $\delta \in \mathbb{R}$ is the allocation of Gaussian element acting the evenness $\delta = w(T\sqrt{loga})$ true. h: $v \to \mathbb{R}$ is a likelihood function of distribution. The statistical gap between the true allocation and the perfect allocation $\frac{2^{-w(loga)}}{a}$ can be determined

by calculating a constant $M$.
Distribution of real:

- $v \rightarrow h$

- $z \rightarrow \mathbb{D}_{\wedge, \delta}, v$

- Outcome $(\lambda, v)$ with likelihood min $\left(\frac{\mathbb{D}_{\wedge, \delta}, v}{a * \mathbb{D}_{\wedge, \delta}, v}, 1\right)$
  Distribution of ideal:

- $v \rightarrow h$

- $\lambda \rightarrow \mathbb{D}_{\wedge, \delta}, v$

- Output $(\lambda, v)$ with likelihood $\frac{1}{a}$

### 3.3 Design goal

To ensure a secure connection in the fog computing with the 5G-assisted vehicular system, privacy and security are both essential. Some privacy and security attributes of the proposed L-CPPA scheme will be thoroughly explained in the information that follows.

- Message Authentication and Integrity: The recipient can accurately confirm the veracity of messages from cars. Recipients can also quickly identify any changes made to the communications they received.

- Identity Privacy-preserving: All vehicles are unable to deduce the true identity of any vehicle from any communication. Additionally, any attacker is unable to determine a vehicle's true identification.

- Traceability and Revocation: When necessary, the messages provided by this vehicle can be used by the TA to trace and revoke any car's true identity.

- Un-linkability: No attacker can link two distinct data coming from the same legitimate car.

- Resistance to Attacks: Our proposed requirement is to withstand various common attacks in the 5G-enabled vehicular fog computing, such as the MITM assault, the modify assault, the replay assault, the forgery assault, and the quantum assault.

### 3.4 Adversary model

The suggested L-CPPA strategy for fog computing with a 5G-assisted vehicular system has an adversary model, the scope and limitations of which are defined in this part. The following is an adversary model that the proposed L-CPPA technique should be able to counter.

- Modify Assaults: A robust policy against impersonation attacks is one that effectively stops attackers from posing as legitimate users.

- Man-In-The-Middle (MITM) Assaults: To resist MITM, a third party must be not able to capture the message sent between the legal senders and receivers.

- Forgery Assaults: To prevent a forgery assaults, the protocol must be able to expose the attacker's effort to fabricate the sent traffic.

- Replay Assaults: To prevent traffic from being recovered from one session and used in another, the protocol should incorporate a nonce or timestamp into each message.

• Quantum Assaults: To resist quantum attacks, traditional cryptography should not be used in the proposed.

## 4 The proposed L-CPPA scheme

In-depth information about the suggested L-CPPA strategy for fog computing with the 5G-assisted vehicular system is presented here. The suggested L-CPPA technique employs both the SampleD [58] and TrapGen [57] crucial algorithms. In addition, the five essential processes of the suggested L-CPPA system are shown in Fig 3: the initialization step, the registration of vehicles, the joining step, the signing of messages, and the verification of signatures. During the registration step, the driver inputs the vehicle's true identity $TID_i$ and password $PWD_i$ to TA for receiving system parameters and verification code. With this step, the vehicle can join the vehicular system to complete the next step as follows. Vehicles hide true identity $TID_i$ by hashing verification codes to obtain anonymous identity $AID_i$, which uses publicly to exchange the message among vehicles or nearby fog servers via 5G-BS. While during message signing,

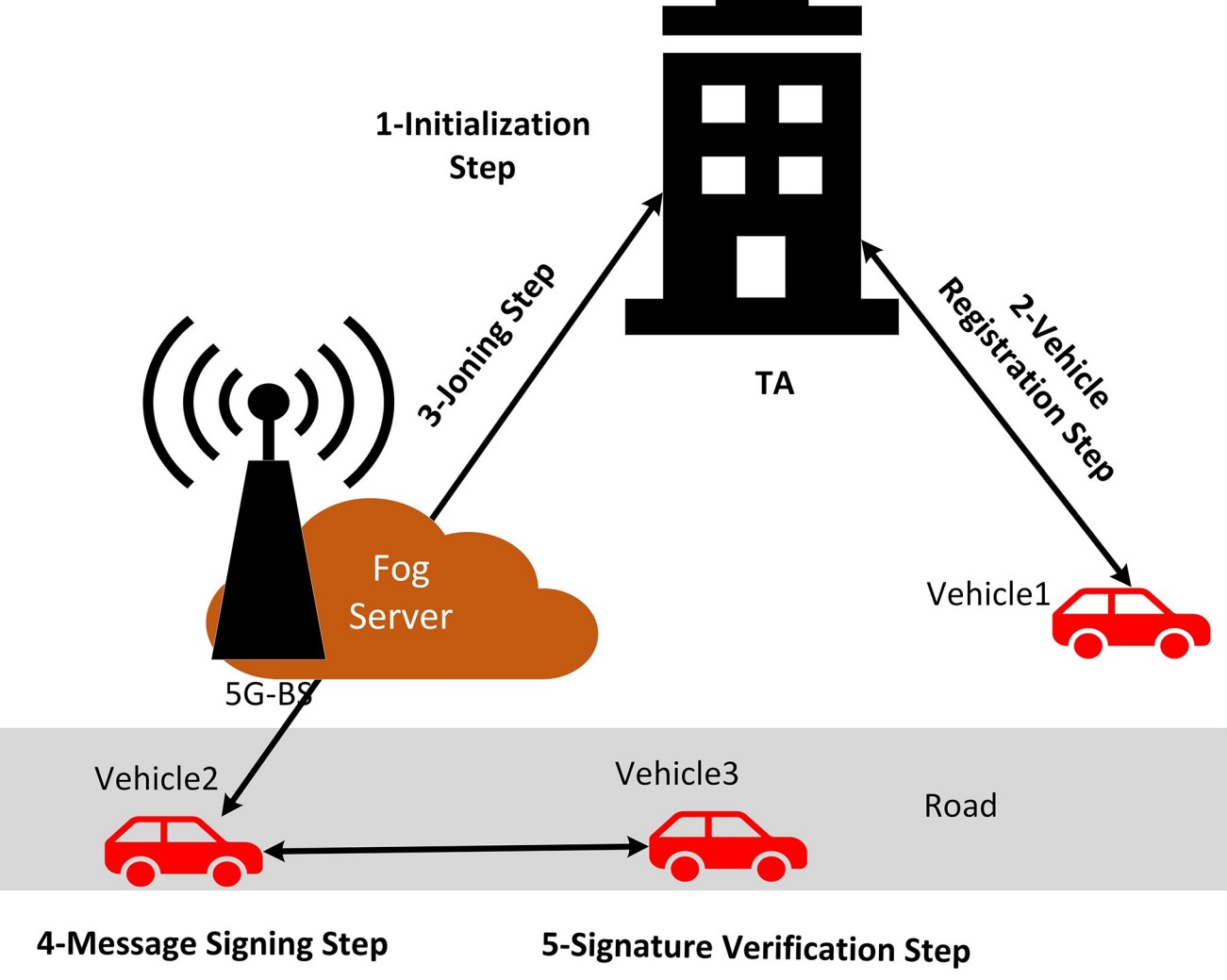

**Fig 3. Steps of the proposed L-CPPA scheme.**

**Table 1. Notation and their description.**

| Notation | Description |
| --- | --- |
| TA | The trusted authority. |
| $fs_j$ | The fog server. |
| $q$ | An odd prime and module. |
| $a, b$ | Two positive integers. |
| $k$ | The security parameter. |
| **E** | The TA's central secret key. |
| **D** | The TA's central public key. |
| $AID_i$ | The i-th vehicle's anonymous identification. |
| $PWD_i$ | The password of the OBU. |
| $TID_i$ | The true identification of the car. |
| $T_i$ | The system's time-stamp, where i = 1, 2, 3, n. |
| $T_\bigtriangledown$ | The delay of predefined time. |
| ‖ | The operation's concatenation. |
| ⊕ | The operation's exclusive-OR. |
| $VC_i$ | The verification code. |
| $h_i$ | The secure general hash functions, where i = 0, 1, 2, 3. |
| $\sigma$ | The signature of message sent |
| $SK_i$ | The vehicle's private key. |
| $SK_i^{enc}$ | The encryption key for private key $SK_i$. |
| $C_i$ | The hash value. |
| $M_i$ | The message exchange among vehicles. |
| $\mu_i, w_i$ | The random values. |
| $Pk_i$ | The signature key. |

the vehicles hide $TID_i$ by hashing the verification code and current timestamp $T_i$ together. Table 1 clarifies the notation applied in this work. $TID_i$ and $AID_i$ refer to true identity and anonymous identity, respectively, which describes in Table 1.

## 4.1 Initialization step

During this procedure, TA releases the system's public and security parameters. Then, TA loads all registered fog servers with the necessary public and security parameters. What follows is a description of that procedure.

- TA picks an item $q > 3$ of odd, security parameter $k$, and two positive integers $a, b > 5k$ $log\ q$.

- TA performs the TrapGen($1^a$, $1^b$, $q$) algorithm to issue the central secret key $\mathbf{E} \in \mathbb{Z}_q^{a \times b}$ and the central public key $\mathbf{D} \in \mathbb{Z}_q^{a \times b}$, where $\mathbf{ED} = 0(mod q)$ and $|\mathbf{E}| \leq 0(a\ log q)$. Additionally, the spread of **D** is vague from a random matrix.

- TA selects four secure general hash operations as follows; $h_0 : \{0, 1\}^* \rightarrow \mathbb{Z}_q^{a \times k}$, $h_1 : \mathbb{Z}_q^{b \times k} \rightarrow \mathbb{Z}_q^a$, $h_2 : \mathbb{Z}_q^a \times \mathbb{Z}_q^a \times \{0, 1\}^* \rightarrow \mathbb{Z}_2^k$, and $h_3 : \mathbb{Z}_q^a \times \mathbb{Z}_q^a \times \{0, 1\}^* \times \mathbb{Z}_q^{a \times k} \times \{0, 1\}^* \rightarrow \mathbb{Z}_q^k$.

- TA sends the parameters of system $\{k, q, a, b, \mathbf{D}, h_0, h_1, h_2, h_3\}$ to all fog servers.

- To ensure the security of all fog servers, TA preloads its master secret key **E**.

## 4.2 Vehicle registration step

Before a car is allowed to leave the factory, it must go through this registration process with TA. What follows is a description of that procedure.

- Driver sends vehicle's true identity $TID_i$ and password $PWD_i$ to TA via a hidden method.

- Upon obtaining the vehicle's identifying information, TA initial confirms the car's legitimacy.

- TA select a random number $VC_i$ as a verification code which uses during the next phase for the first verification with the fog server. Then TA preloads $VC_i$ to the TPD of OBUs.

- TA broadcasts the system parameters $\{k, q, a, b, \mathbf{D}, h_0, h_1, h_2, h_3\}$ to all OBUs.

- TA saves tuples $\{TID_i, PWD_i, VC_i\}$ on the list of vehicle registration.

## 4.3 Joining step

To begin using the car as an legalized/enrolled node in a 5G-assisted vehicular system with fog computing, the car, the fog server, and the TA all need to carry out this step. The steps involved are outlined below.

- The anonymous pair identifier $AID_i$ is calculated as follows: vehicle $v_i$ uses verification code ($VC_i$) and chooses number $w_i$.

$$AID_i = AID_i^1, AID_i^2$$
$$= w_i \cdot \mathbf{D}, TID_i \oplus h_1(VC_i \| T_1) \tag{6}$$

- Vehicle $v_i$ then transmits items $\{AID_i^1, AID_i^2, T_1, \sigma_1\}$ to fog server $fs_j$ via communication zone by 5G-BS, where $T_i$ is freshness timestamp and $\sigma_1 = h_2(AID_i^1 \| AID_i^2 \| T_1)$.

- When receiving tuples $\{AID_i^1, AID_i^2, T_1, \sigma_1\}$, fog server $fs_j$ confirms the tenderness of time-stamp in order to avoid replay attacks as the following equation.

- Fog server $fs_j$ checks the integrity of the tuple by computing the following equation.

$$\sigma_1 \stackrel{?}{=} h_2(AID_i^1 \| AID_i^2 \| T_1) \tag{7}$$

If Eq (7) doesn't hold, fog server $fs_j$ rejects for completing joining steps since the attacker has occurred. Otherwise, fog server $fs_j$ continues the next steps.

- Fog server $fs_j$ sends tuples $\{AID_i^1, AID_i^2, T_1, \sigma_1\}$ to TA in order to check the authenticity of the vehicle.

- When receiving tuples $\{AID_i^1, AID_i^2, T_1, \sigma_1\}$, TA extracts car's authentic identification by computing $TID_i = AID_i^2 \oplus h_1(VC_i)$.

- TA checks the authenticity of the car's authentic identification $TID_i$ whether matching in the list of vehicle registration or not. If it is not ok, TA transmits $illegal_{node}$ to fog server $fs_j$. Otherwise, TA transmits ($legal_{node}$, $VC_i$, $TID_i$) to fog server.

- When receiving $illegal_{node}$, fog server $fs_j$ discards the tuple $\{AID_i^1, AID_i^2, T_1, \sigma_1\}$ and ends the session. Otherwise, fog server $fs_j$ continues the next steps.

- Fog server $fs_j$ computes the hash number $C_i = h_0(TID_i)$ and selects a true value $\delta \geq \|\mathbf{E}' * a(\sqrt{logb})\|$, where $\mathbf{E}'$ indicates the orthogonalization of the matrix $\mathbf{E}$.

- Fog server $fs_j$ runs the algorithm of SampleD($\mathbf{D}$, $\mathbf{E}$, $\delta$, $C_i$) to produce secret key $SK_i \in \mathbb{Z}_q^{b \times k}$, where $\mathbf{D} * SK_i = C_i(modq)$.

- Fog server $fs_j$ encrypts $SK_i^{enc} = SK_i \oplus h_1(VC_i)$ and computes $\sigma_2 = h_2(SK_i^{enc} \| T_2)$. Fog server $fs_j$ then sends tuples $\{SK_i^{enc}, T_2, \sigma_2\}$ to vehicle $v_i$.

- When receiving tuples $\{SK_i^{enc}, T_2, \sigma_2\}$ from fog server $fs_j$, vehicle $v_i$ tests newness of timestamp $T_2$ in order to resist replay attacks.

- Vehicle $v_i$ decrypts private key $SK_i^{enc}$ by computing $SK_i = SK_i^{enc} \oplus h_1(VC_i)$. Vehicle $v_i$ then confirms the originality and integrity of tuples by computing the following equation.

$$\sigma_2 \overset{?}{=} h_2(SK_i^{enc} \| T_2) \tag{8}$$

If Eq (8) doesn't hold, vehicle $v_i$ rejects for completing joining steps since the attacker has occurred. Otherwise, vehicle $v_i$ saves its private key $SK_i$ on TPD to continue the next steps.

Note that the way to check the freshness of timestamp $T_i$ by computing Eq 9, where $T_r$ is a delay of receiving time and $T_{\bigtriangledown}$ is a delay of predefined time.

$$T_i > T_r - T_{\bigtriangledown} \tag{9}$$

## 4.4 Message signing step

In this step, the car $v_i$ that wants to relay message $M_i$ to the other vehicles of fog servers does so. This procedure is explained below.

- In order to accomplish the unlinkability characteristic, vehicle $v_i$ chooses a value $\mu_i$ and uses the following equation, where $T_i$ is the existing device timestamp, to compute the pair anonymous identity $AID_i$.

$$\begin{aligned} AID_i &= AID_i^1, AID_i^2 \\ AID_i^1 &= \mu_i \cdot \mathbf{D}, \\ AID_i^2 &= TID_i \oplus h_1(VC_i \| T_i) \end{aligned} \tag{10}$$

- Vehicle $v_i$ calculates the signature key by generating $Pk_i = \mu_i + SK_i \cdot h_2(AID_i^1 \| AID_i^2 \| T_i)$

- Vehicle $v_i$ selects a random matrix $\mathbf{A}_i \in \mathbb{Z}_q^{b \times k}$ to compute $\mathbf{B}_i = \mathbf{A}_i \cdot \mathbf{D}$.

- Vehicle $v_i$ computes hash value signature $\sigma_i = Pk_i + \mathbf{A}_i \cdot h_3(M_i \| AID_i^1 \| AID_i^2 \| T_i \| \mathbf{B}_i)$ for message $M_i$.

- Finally, vehicle $v_i$ broadcasts $\{M_i, \mathbf{B}_i, AID_i, T_i, \sigma_i\}$ to others.

## 4.5 Signature verification step

In this step, the recipient (a moving vehicle or a fog server) is validated to ensure the signature it has received is genuine. This method can be used for either single-check or batch-check verification. The following section elaborates on these steps.

**4.5.1 Single verification process.** Here, the suggested L-CPPA technique only requires one verification step to ensure that the signature $\{AID_1, M_1, \mathbf{B}_1, T_1, \sigma_1\}$ was sent by the intended sender, vehicle $v_1$. What follows is an explanation of how this works.

- After receiving $\{AID_1, M_1, \mathbf{B}_1, T_1, \sigma_1\}$, the verifier checks first freshness of $T_1$, the verifier discards to do the coming step.

- The verifier then tests whether Eq 11 holds.

$$
\begin{aligned}
\sigma_i \cdot \mathbf{D} &\stackrel{?}{=} \left( Pk_i + \mathbf{A}_i \cdot h_3(M_i\|AID_i^1\|AID_i^2\|T_i\|\mathbf{B}_i) \right) \cdot \mathbf{D} \\[6pt]
&\stackrel{?}{=} \Big( (\mu_i + SK_i \cdot h_2(AID_i^1\|AID_i^2\|T_i)) + \\[6pt]
&\quad \mathbf{A}_i \cdot h_3(M_i\|AID_i^1\|AID_i^2\|T_i\|\mathbf{B}_i) \Big) \cdot \mathbf{D} \\[6pt]
&\stackrel{?}{=} (\mu_i \cdot \mathbf{D} + h_2(AID_i^1\|AID_i^2\|T_i) \cdot SK_i \cdot \mathbf{D}) + \\[6pt]
&\quad (h_3(M_i\|AID_i^1\|AID_i^2\|T_i\|\mathbf{B}_i) \cdot \mathbf{A}_i \cdot \mathbf{D}) \\[6pt]
&\stackrel{?}{=} AID_i^1 + h_2(AID_i^1\|AID_i^2\|T_i) \cdot h_0(TID_i) + \\[6pt]
&\quad h_3(M_i\|AID_i^1\|AID_i^2\|T_i\|\mathbf{B}_i) \cdot \mathbf{B}_i
\end{aligned}
\tag{11}
$$

If the checker is unable to successfully check Eq 11, then the notification $M_i$ may be ignored. If $\sigma_i$ is not genuine, the verifier will not accept the letter $M_i$.

**4.5.2 Batch verification process.** The suggested L-CPPA system might use batch verification mode to check the authenticity and integrity of many signatures at once, which is an improvement above the efficiency of traditional verification (e.g., a single verification operation). What follows is a description of that procedure.

- After receiving multiple signatures $\{AID_1, M_1, \mathbf{B}_1, T_1, \sigma_1\}, \{AID_2, M_2, \mathbf{B}_2, T_2, \sigma_2\}, \ldots, \{AID_n, M_n, \mathbf{B}_n, T_n, \sigma_n\}$, the verifier checks first freshness of $T_i \in [n]$. The verifier won't perform the subsequent operation if one of those timestamps cannot be satisfactorily confirmed.

- The verifier chooses random small numbers $\{\gamma_1, \gamma_2, \ldots, \gamma_n\}$, where $\gamma_i \in \mathbb{Z}_q$.

- The verifier then confrims whether Eq 12 holds or not:

$$
\begin{aligned}
\mathbf{D} \cdot \left( \sum_{i=1}^{n} \gamma_i \cdot \sigma_i \right) &\stackrel{?}{=} \sum_{i=1}^{n} \gamma_i \sigma_i \cdot \mathbf{D} \\[6pt]
&\stackrel{?}{=} \sum_{i=1}^{n} \gamma_i \cdot AID_i^1 + \sum_{i=1}^{n} \gamma_i h_2(AID_i^1\|AID_i^2\|T_i) \cdot h_0(TID_i) \\[6pt]
&\quad + \sum_{i=1}^{n} \gamma_i h_3(M_i\|AID_i^1\|AID_i^2\|T_i\|\mathbf{B}_i) \cdot \mathbf{B}_i
\end{aligned}
\tag{12}
$$

If Eq 12 is verified, the checker believes all values $\sigma_i$, $i \in [n]$ are legal and doesn't reject the corresponding data $M_i$, $i \in [n]$. The opposite, the checker rejects these information $Msg_i$, $i \in [n]$.

## 5 Provable security

This section proves the security model, security analysis and security attributes of the suggested L-CPPA system for a 5G-assisted vehicular system with fog computing.

### 5.1 Security model

This subsection describes the security model of the proposed L-CPPA scheme based on the ability of the adversary and the network model of the 5G-assisted vehicular fog computing. The security model is essentially a safeguard test that adversary $A$ and challenger $C$ are playing. The following inquiries may be made in order to help $A$ acquire the necessary skills for the security game.

- Oracle $(h_0)$: The main idea of this query is the challenger $C$ selects a random matrix $\boldsymbol{X_i} \, \mathbb{Z}_q^{n \times k}$ and adds the tuple $(TID_i, \boldsymbol{X_i})$ into the $List_{h_0}$. $C$ returns the matrix $\boldsymbol{X_i}$ to $A$.

- Oracle $(h_1)$: Once obtaining the query from $A$, $C$ selects a random value $v_j \in \mathbb{Z}_q^n$ and puts the tuple $(Sk_i, v_j)$ into the $List_{h_1}$. $C$ returns the value $v_j$ to $A$.

- Oracle $(h_2)$: Once obtaining $\{AID_i, T_i\}$ from $A$, $C$ selects a random value $y_j \in 0, 1^k$ as the answer outcome and gives $y_j$ to $A$. $C$ puts the tuple $\{AID_i, T_i, y_j\}$ into the $List_{h_2}$.

- Oracle $(h_3)$: In this query, $C$ picks a random item $z_i \in \mathbb{Z}_q^k$ as the response result of the tuple $\{AID_i, T_i, B_i, M_i\}$. Additionally, $C$ puts the query element $\{AID_i, T_i, B_i, M_i, z_i\}$ into the $List_{h_3}$.

- Oracle $(sign)$: In this query, $A$ sends a message $M_i$ to $C$. The query output is that $C$ computes a tuple $\{AID_i, M_i, \mathbf{B}_i, T_i, \sigma_i\}$ and gives this tuple to $A$.

### 5.2 Security analysis

**Theorem 1**: In the random oracle model, the proposed L-CPPA technique is safe against an adversary with polynomial time complexity $A$, as measured by the hardness of SIS/ISIS issues.

 **Proof**: To demonstrate this theorem, C and A engage in what amounts to a security game. In this game's specific procedure, opponent $A$ is assumed to have the ability to make Oracle $(h_0)$ queries $Qh_0$ times, $Qh_1$ queries $Qh_1$ times, $Qh_2$ queries $Qh2$ times, $Qh3$ queries $Qh_3$ times, and $Qs$ queries $Qs$ times. It also assumes that $A$ can compromise our scheme's existential unforgeability with a probability of *varepsilon*. In addition, it presupposes that $TID_i$ is the identity of the vehicle associated with the signature that $A$ wants to fake. Using the knowledge $(D)$, and the power $(U)$, $C$ hopes to defeat ISIS and win the game. In this security game, the challenger $C$ and the adversary $A$ specifically do the below:

- Setup: Once obtained the safeguard element $n$, $C$ produces the central public key $\mathbf{D}$ using the system parameters $\{m, k, q, h_0, h_1, h_2, h_3\}$. After that, $C$ sends $A$ the updated settings for $B$.

- Query: In the security test, the attacker could use five distinct Oracle queries: Oracle $(h_0)$ query, Oracle $(h_1)$ query, Oracle $(h_2)$ query, Oracle $(h_3)$ query, and Oracle-based Sign $(sign)$ query. The $C$ also keeps track of the elements of these hash oracle queries in four empty lists: $L_{h_0}, L_{h_1}, L_{h_2}$, and $L_{h_3}$. As a next stage, $C$ will re-run these questions.

  - $(h_0)$: Once obtainin the query $\{TID_i\}$ from $C$, $A$ initial confirms if the tuple $(TID_i, \boldsymbol{X_i})$ has existed in the list $L_{h_0}$ or not. If this element is in $L_{h_0}$, $C$ gives $\boldsymbol{X_i}$ to the adversary. Otherwise, $C$ random selects a matrix $\boldsymbol{X_i} \, \mathbb{Z}_q^{n \times k}$ and puts the element $(TID_i, \boldsymbol{X_i})$ into the list $L_{h_0}$. Finally,

$C$ outputs this attacker's query with $X_i$. Especially, $C$ replays this query with $U$ if $TID_i = TID_i^*$.

- $(h_1)$: In response to a query $Sk_i$ sent from $A$, $C$ will first locate the entire list $L_{h_1}$ for $(Sk_i, v_i)$. Whenever this component is present, $C$ will provide $v_i$ to $A$. If not, $C$ will respond with a random vector value, $v_j \in \mathbb{Z}_q^n$. The item $(Sk_i, v_i)$ will then be inputted to the list $L_{h_1}$.

- $(h_2)$: If $\{AID_i, T_i\}$ is an Oracle query conducted by $A$, then $C$ will determine if it already exists. A is given $yj$ by $C$ from $L_{h_2}$ if and only if the pair $\{AID_i, T_i, y_j\}$ has already appeared in that set. If not, $C$ chooses a random vector value from $\{AID_i, T_i, y_j\}$ and sends it back to $A$ as the value $y_j$. And finally, C appends the item $\{AID_i, T_i, y_j\}$ to the list $L_{h_2}$.

- $(h_3)$: After $A$'s query $\{AID_i, T_i, B_i, M_i\}$ returns results, $C$ first chooses whether or not the item in question has been added to the list $Lh3$. The value $zi$ is returned by $C$ from this query if and only if the element $\{AID_i, T_i, B_i, M_i, z_i\}$ exists. In any other case, $C$ picks a random element from $z_i \in \mathbb{Z}_q^k$ and appends the element $\{AID_i, T_i, B_i, M_i, z_i\}$ to the list $L_{h_3}$. After everything is said and done, $z_i$ is forwarded to A.

- (*Sign*): With *Sign* process, $C$ does not realize any data around the central private key $\mathbf{E}$ and any vehicle $TID_i$'s secret key $Sk_i$. Thus, $C$ can not opportunely answer a sign query from $A$ by running the data signing step. If $M_i$ is the oracle (**Sign**) query made by $A$, $C$ first random selects $\sigma_i$ and $AID_i^2$. Then $C$ random selects $\alpha_i$, $\Upsilon_i$, and $B_i$. After, $C$ can calculates $AID_i^1 = \mathbf{D} \cdot \sigma_i - h_0(TID_i) \cdot \alpha_i + Z_i \cdot \Upsilon_i(mod)$. After that, $C$ inputs $AID_i^1$ and $AID_i^2$. Finally, $C$ gives the tuple $\{AID_i, M_i, \mathbf{B}_i, T_i, \sigma_i\}$ to the adversary $A$ as the query output.
  An adversary $A$ can verify *sigmai*'s authenticity after acquiring the tuple $\{AID_i, M_i, \mathbf{B}_i, T_i, \sigma_i\}$ from $C$. However, you'll need to pick out $Event_1$ and $Event_2$. To execute the oracle (Signing) query, $A$ must first execute the $Event1$ event of running an oracle $h2$ query such that $\alpha_i = h_2(AID_i, T_i)$. When $A$ runs the oracle $h3$ query, $Event_2$ occurs because v. This occurs before the oracle (Signing) query. Challenger $C$ loses the security game if $Event1$ and $Event2$ occur.

- Forgery: If $C$ has completed the Setup and Query phases successfully, then $A$ will be chosen to produce a legal signature in this step. The tuple $\{AID_i, M_i, \mathbf{B}_i, T_i, \sigma_i\}$ is considered $A$'s signature since we presume that $A$ can violate the empirical unforgeability of our proposal. It has $\sigma_i^* \cdot \mathbf{D} = AID_i^1 + \alpha_i^* \cdot h_0(TID_i) + \Upsilon_i^* \cdot \mathbf{B}_i$. Where $\alpha_i^* = h_2(AID_i^1 \| AID_i^2 \| T_i)$ and $\Upsilon_i^* = h_3(M_i \| AID_i^1 \| AID_i^2 \| T_i \| \mathbf{B}_i)$.

Based on the forking lemma [60], $C$ reassigns $h_2(AID_i^* \| T_i^*)$ to $\alpha_i''$. The game was then played again by $C$ with $A$ as the adversary. Likewise, $A$ can create a legal signature at the end of another test. For simplicity, it supposes that the tuple $\{AID_i^*, M_i^*, \mathbf{B}_i^*, T_i^*, \sigma_i^*\}$ is the the signature executed by A. Thus, it has $\sigma_i^- \cdot \mathbf{D} = AID_i^{1,*} + \alpha_i'' \cdot h_0(TID_i) + \Upsilon_i^* \cdot \mathbf{B}_i^*$. When the aforementioned two values could be correctly validated, $C$ can figure out the equation shown below:

$$\mathbf{A}(\sigma_i^* - \sigma_i') = h_0(TID_i)(\alpha_i^* - \alpha_i').$$

With applying the above values, $C$ could calculate the result of the challenge of ISIS. We select an element $\lambda$ for instance $(\alpha_i^* - \alpha_i')\lambda = \mathbf{I}^{k \cdot k} \bmod q$, where $\mathbf{I}$ is the identification matrix. Therefore, we define that

$$\mathbf{A}(\sigma_i^* - \sigma_i')\lambda = h_0(TID_i).$$

Since $\alpha_i^*$ and $\alpha_i'$ are both various values and $\lambda$ is an element of non-zero, $C$ could utilises $A$ to resolve the challenge of ISIS passing. Additionally, the likelihood of $C$ to resolve the challenge of ISIS is:

$$
\begin{aligned}
Pr[C - pass] \quad &= Pr[A - pass | Event_1^* \wedge Event_2^*] \\
&= \frac{Pr[A - pass | Event_1^* \wedge Event_2^*]}{Pr[Event_1^*] + Pr[Event_2^*]} \\
&\leq \frac{\varepsilon}{1 - Pr[Event_1] + 1 - Pr[Event_2]} \\
&\leq \frac{\varepsilon}{1 - \frac{Qh_2}{2^{2n}} + 1 - \frac{Qh_3}{2^{4n}}} \\
&\leq \frac{\varepsilon}{2 - \frac{2^{2n}Qh_2 + Qh_3}{2^{4n}}} \\
&\leq \frac{2^{4n}}{2^{4n+1} - (2^{2n}Qh_2 + Qh_3)}\varepsilon
\end{aligned}
\tag{13}
$$

Nevertheless, as is common knowledge, no polynomial-time algorithm can resolve the problem of ISIS. The proposed L-CPPA scheme is safe, as per the notion of proof by contradiction.

## 5.3 Security attributes

This section discusses the security attributes of the suggested L-CPPA system required to achieve as follows.

- **Theorem 2**: The suggested L-CPPA system for fog computing with a 5G-assisted vehicular system achieves message authenticity and integrity.
  **Proof**: From **Theorem 1**, it can conclude which a legal value can not be faked by any attacker based on time based polynomial since the ISIS problem difficulty. Thus, the verifying recipient has capable to verify the authenticity and validity of a legal signature $\{AID_i, M_i, \mathbf{B}_i, T_i, \sigma_i\}$ easily by checking Eqs 11 and 12. Therefore, the suggested L-CPPA system can realize message integrity and authenticity.

- **Theorem 3**: The suggested L-CPPA systemfor fog computing with 5G-assisted vehicular system achieves identification privacy-preserving.
  **Proof**: A vehicle's true identification $TID_i$ is utilised to create its anonymity-IDs $AID_i$ by its OBU. We define that $AID_i^1 = \mu_i \cdot \mathbf{D}$, $AID_i^2 = TID_i \oplus h_1(VC_i \| T_i)$, and $AID_i = AID_i^1, AID_i^2$. After obtaining this anonymity $AID_i$, an attacker should have the sensitive element $Sk_i$ or resolve the another value of $h_1(Sk_i)$ if it wishes to reveal $TID_i$ from $AID_i$. However, there is very little chance that one of these events will occur. As a result, the suggested L-CPPA system can safeguard the anonymity of individual identities within fog computing with a 5G-assisted vehicular system.

- **Theorem 4**: The suggested L-CPPA system for fog computing with 5G-assisted vehicular system achieves traceability.
  **Proof**: In fog computing with a 5G-assisted vehicular system, the message is exchanged among vehicles by using their anonymity-IDs. The anonymity-IDs $AID_i$ of a car is issued by its OBU utilising $TID_i$ and the relevant secret key $Sk_i$. We know that $AID_i = AID_i^1, AID_i^2$,

and $AID_i^2 = TID_i \oplus h_1(VC_i \| T_i)$. If required, TA saves value $VC_i$ during the registration list. Then, TA can easily reveal $TID$ from $AID_i^2$ by calculating $TID_i = AID_i^2 \oplus h_1(VC_i \| T_i)$. So the proposed L-CPPA scheme can achieve traceability of a vehicle's authentic identification in fog computing with a 5G-assisted vehicular system.

- **Theorem 4**: The suggested L-CPPA system for fog computing with 5G-assisted vehicular system achieves un-linkability.

  **Proof**: To create a legal parameters $\{AID_i, M_i, \mathbf{B}_i, T_i, \sigma_i\}$ in the fog computing with 5G-assisted vehicular system, a vehicle random selects an element $\mu_i$ and a matrix $\mathbf{A}_i$ during running the proposed L-CPPA scheme. We define that $Pk_i = \mu_i + SK_i \cdot h_2(AID_i^1 \| AID_i^2 \| T_i)$ and $\sigma_i = Pk_i + \mathbf{A}_i \cdot h_3(M_i \| AID_i^1 \| AID_i^2 \| T_i \| \mathbf{B}_i)$. Owing to the randomness of $\mu_i$ and $\mathbf{A}_i$, an adversary with a polynomial time limit cannot link any two various anonymous identification or signatures from the same source. As a result, fog computing with the 5G-assisted vehicular system can achieve unlinkability using the proposed L-CPPA scheme.

- **Theorem 5**: The suggested L-CPPA system for fog computing with a 5G-assisted vehicular system resists lots of attacks, such as the modify assault, the MITM assault, the forgery assault, the replay assault, and the quantum assault.

  - Modify Assault: In the suggested L-CPPA system, any alteration on the tuple $\{AID_i, M_i, \mathbf{B}_i, T_i, \sigma_i\}$ can be founded by checking Eqs 11 and 12. Thus, the modify assault could be withstood by the suggested L-CPPA scheme.

  - MITM Assault: The suggested L-CPPA system can implement message authentication between two vehicles in fog computing with a 5G-assisted vehicular system, according to **Theorem 2**. As a result, our suggested is resistant to a MITM assault.

  - Forgery Assault: The adversary must successfully forge a legitimate signature $\{AID_i, M_i, \mathbf{B}_i, T_i, \sigma_i\}$ on a message $M_i$ in order to pass as a vehicle. Namely, Eqs 11 and 12 must be achieved. Nevertheless, Due to the difficulty of the ISIS problem, the benefit of the adversary producing such a data tuple is quite small. Consequently, the suggested L-CPPA system can resist a forgery assault.

  - Replay Assault: In the suggested L-CPPA scheme, a time-stamp $T_i$ is inserted in a value $\{AID_i, M_i, \mathbf{B}_i, T_i, \sigma_i\}$. Once testing this signature, fog servers and other vehicles would test the newness of $T_i$. Therefore, the time-stamp $T_i$ achieves the suggested L-CPPA scheme to resist the replay assault.

  - Quantum Assault: The unforgeability-based existential of the suggested system is founded on the ISIS/SIS problems, according to the security analysis. However, the ISIS/SIS problems cannot be resolved by the quantum adversary. We would conclude that the L-CPPA scheme under consideration is immune to quantum assault.

## 6 Performance evaluation and comparison

This part evaluates and compare the suggested L-CPPA system and two other authentication schemes [53, 54] in terms of computation and communication costs.

Due to the problem of ISIS forming the basis for the security of the suggested L-CPPA system in 5G-assisted vehicular fog computing, we carefully choose the pertinent parameters to ensure that the ISIS challenge achieves the desired safeguard standard. Our scheme's module $q$ was set to 101 in order to achieve the safeguard standard of 123 bits. Moreover, the master public key's number $\mathbf{D}_i \in \mathbb{Z}_q^{a \times b}$ of rows and columns are 100 and 666. More specific, $a = 100$

and $b = 666$. The secret parameter $k$ for an OBU has 80 columns. In addition, the discrete Gaussian distribution in our procedure has a standard deviation $\delta$ of 1.0038.

## 6.1 Computation costs

In this part, we conduct a thorough evaluation of our scheme's computational overhead. This cost of the suggested L-CPPA system and two other works [53, 54] is compared. Table 2 lists some notations used and their description and execution in this section. This paper uses an experiment in [55] that uses a cryptographic library called NTL, which is a familiar numbering theory library. On a hardware platform with an Intel(R) Core(TM) i5-10700 CPU operating at 2.9 GHz, 4 GB of RAM, and Windows 10 for operation (64-bit), the suggested L-CPPA system is implemented.

For simplicity, *MSP*, *SVP*, and *BVP* indicate the message signing step, single verification process, and batch verification process, respectively. In the *MSP* for a scheme of Mukherjee et al. [53], a component only requires executing a keyed-hash message authentication process and a secure hash function operation. Entire computation cost is $T_{hash} + T_{mac} \approx 0.0227$ ms. In *SVP*, the component requires to run a keyed-hash data authentication process and a hash function operation. Entire computation cost is $T_{hash} + T_{mac} \approx 0.0227 ms$. As for the *BVP*, the scheme of Mukherjee et al. [53] does not satisfy the batch verification of numerous data.

In the *MSP* for the scheme of Dharminder et al. [54], two vector multiplications with integers, two matrix multiplications with vectors, three hash operations, one XOR operation, and two vector additions in $\mathbb{Z}_q^a$ are needed to build a car. Additionally, an OBU requires sampling a vector from $\mathbb{Z}_q^a$. Entire computation cost is $T_{sp} + 2T_{va}^{ba} + T_{xor} + 3T_{hash} + 2T_{vec}^{ala} + 2T_{vec}^{saa} \approx 113.183 ms$. In

**Table 2. Some notations used and their description and execution.**

| Notation | Description | Execution Time |
|---|---|---|
| $T_{vec}^{sak}$ | The running time takes to complete adding items in $\mathbb{Z}_q$. Each of those vectors has a dimension of $k$. | 0.47 ms |
| $T_{vec}^{sab}$ | The running time takes to complete adding items in $\mathbb{Z}_q$. Each of those vectors has a dimension of $b$. | 0.059 ms |
| $T_{vec}^{saa}$ | The running cost takes to complete adding items in $\mathbb{Z}_q$. Each of those vectors has a dimension of $a$. | 0.385 ms |
| $T_{vec}^{ala}$ | Suppose that $x$ is a items in $\mathbb{Z}_q^a$ and $m \in \mathbb{Z}_q$. $T_{vec}^{ala}$ is the running cost of m * x mod q. | 0.697 ms |
| $T_{vec}^{alb}$ | Suppose that $y$ is a items in $\mathbb{Z}_q^n$ and and $n \in \mathbb{Z}_q$. $T_{vec}^{alb}$ is the running cost of y * a mod q. | 0.074 ms |
| $T_{va}^{ab}$ | The cost required for the procedure of multiplying a matrix by an element on a mod q to complete. Moreover, this matrix has $a$ rows and $b$ columns. | 48 ms |
| $T_{va}^{ak}$ | Suppose that $D$ is item in $\mathbb{Z}_q^{a*k}$ and $x$ is a item in $\mathbb{Z}_q^k$. $T_{va}^{ak}$ is the running cost of the multiplication **D**x mod q. | 28 ms |
| $T_{va}^{bk}$ | Suppose that $D$ is item in $\mathbb{Z}_q^{b*k}$ and $y$ is an element in $\mathbb{Z}_q^k$. $T_{va}^{bk}$ is the running cost of the multiplication **D**y mod q. | 6 ms |
| $T_{mat}^{mul}$ | The running costs of the multiplication of factories on module q is indicated by $T_{mat}^{mul}$. Additionally, the prime matrix is an item in $\mathbb{Z}_q^{a*b}$ and the other matrix is an item in $\mathbb{Z}_q^{b*k}$. | 419 ms |
| $T_{hash}$ | The execution cost of a hash function is indicated by this symbol. The running times of $h_1$, $h_2$, and $h_3$ in the proposed L-CPPA scheme can be accurately reflected by this execution time, it is important to note. Since the hash function $h_0$ can be calculated offline, its running time can be disregarded. | 0.006 ms |
| $T_{xor}$ | The running cost of XOR process of both elements in $\mathbb{Z}_q^b$. | 0.0002 ms |
| $T_{sp}$ | The duration of selecting a random vector with a discrete Gaussian distribution. | 15 ms |

**Table 3. Summary of computational cost comparison.**

| Approach | MSP | SVP | BVP |
|---|---|---|---|
| Mukherjee et al. [53] | $T_{hash} + T_{mac} \approx 0.0227ms$ | $T_{hash} + T_{mac} \approx 0.0227ms$ | NULL |
| Dharminder et al. [54] | $T_{sp} + 2T_{va}^{ba} + T_{xor} + 3T_{hash} + 2T_{vec}^{ala} + 2T_{vec}^{saa} \approx 113.183ms$ | $T_{va}^{ba} + 2T_{vec}^{ala} + 2T_{vec}^{saa} \approx 50.164ms$ | $T_{va}^{ba} + (3N+1)T_{vec}^{ala} + 2N*T_{vec}^{saa} \approx 334.797ms$ |
| Proposed L-CPPA Scheme | $(2e+1)T_{hash} + e*(T_{sp} + T_{xor} + T_{va}^{ab}) + (e+1)(T_{vec}^{saa} + T_{va}^{ak}) + T_{mal}^{mul} \approx 694.161ms$ | $T_{va}^{ba} + 2*T_{vec}^{sab} \approx 60.118ms$ | $2N*T_{va}^{bk} + 3N*T_{vec}^{alb} + N*T_{vec}^{ala} + T_{va}^{ba} + (3N+2)T_{vec}^{sba} + N*T_{vec}^{sba} \approx 1348.218ms$ |

*SVP*, to operate on the vehicle, two vector multiplications and an integer addition must be performed in $\mathbb{Z}_q^a$, a single operation consisting of multiplying a matrix by a vector. Entire computation cost is $T_{va}^{ba} + 2T_{vec}^{ala} + 2T_{vec}^{saa} \approx 50.164ms$. As for the *BVP*, the vehicle requires to run single multiplication procedure of an element and matrix, 2N addition procedures of vectors in $\mathbb{Z}_q^a$, 3N + 1 multiplication procedures of an integer and a vector. Entire computation cost is $T_{va}^{ba} + (3N+1)T_{vec}^{ala} + 2N*T_{vec}^{saa} \approx 334.797ms$. The same method to compute the computation costs of the suggested L-CPPA system in terms of *MSP*, *SVP*, and *BVP*. Table 3 lists the precise computation costs for each of the three authentication and privacy-preserving schemes procedures.

## 6.2 Costs of communication

This section shows the communicational overheads of the proposed L-CPPA scheme and the relevant studies of Mukherjee et al. [53] and Dharminder et al. [54]. For simplicity, the size of a timestamp must be assumed to be 4 bytes, and all authentication schemes require the same amount of data to sign messages. Additionally, the other two schemes have the same b, a, k, q system parameters as our scheme. Thus, only the size of the signatures produced by these authentication systems needs to be taken into account.

In the scheme of Mukherjee et al. [53], the node transmits the final signature $\{SPID_i^k, ts, \delta_i\}$ to all participating nodes in the system. The size of *ts*, $SPID_i^k$ and the signature are 20 bytes, 20 bytes and 64 bytes, respectively. Additionally, the $\delta_i$ contains two 20-byte hash amounts.

In scheme of Dharminder et al. [54], the tuples $\{R_i, SK_i, T_i, AID_i\}$ sent to the node. Therefore, the quantity of this signature is nearby 4 + 846 = 6800 bits. The quantity of $AID_i$ is 2 * n * $log_2$ (q-1) = 175 bytes. Additionally, the quantity of $R_i$ is $nlog_2(q1) = 87.5$ bytes and that of $Sk_i$ is $mlog_2(q1) = 4662$ bits.

In the proposed L-CPPA scheme, the vehicle transmits the final signature $\{AID_i, M_i, \mathbf{B}_i, T_i, \sigma_i\}$ to other vehicles and fog servers. Since $AID_i = (AID_i^1, AID_i^2)$ and $AID_i^2 = TID_i \oplus h_1(VC_i\|T_i)$, $AID_i^2$ should be a random item in $\mathbb{Z}_q^b$. To top the quantity of $AID_i^2$, we let all amounts of $AID_i^2$ be $q-1$. Likewise, $AID_i^1$ is a random item in $\mathbb{Z}_q^b$. It could readily compute the major size of $AID_i$, which is $2nlog_2(q1) = 1400$ bits. Since $\mathbf{B}_i = \mathbf{A}_i \cdot \mathbf{D}$ and $\mathbf{A}_i \in 0, 1^{a*k}$, the top item of $\mathbf{B}_i$ is $q-1 = 7000$ bytes. Since of $\sigma_i \in \mathbb{Z}_q^b$, the top of this element is $q-1$. Thus, the costs of $\sigma_i$ is $b * log_2(q-1) = 583$ bytes. The quantity of the tuples $\{AID_i, M_i, \mathbf{B}_i, T_i, \sigma_i\}$ is approximately 7757 bytes. Table 4 tabulates the comparison of the communicational overheads of the suggested L-CPPA system and the other two schemes.

**Table 4. Summary of communication cost comparison.**

| Study | Message-Signature Pair | N Pair |
|---|---|---|
| Mukherjee et al. [53] | 64 bytes | N*64 bytes |
| Dharminder et al. [54] | 850 bytes | N*850 bytes |
| L-CPPA | 7757 bytes | N*7757 bytes |

Despite having a substantially higher communication cost than the other two schemes, the suggested L-CPPA system's signature size is still suitable for a 5G-assisted vehicular system with fog computing. because the proposed scheme's signature is a reasonable size for the 5G-BS network.

## 7 Conclusions

This work has suggested an L-CPPA system in order to realize piracy and security attributes for 5G-assisted vehicular systems with fog computing. Fog servers are used in the planned L-CPPA concept to generate and transmit secret keys to each registered car across 5G-BS. In the proposed L-CPPA approach, the trusted authority stores its master private key to each fog server rather than the OBU of the vehicle. A thorough security study and formal security proof under ROM have demonstrated that the protocol provides adequate protection for message authenticity and integrity, unlinkability, traceability, identification privacy-preserving, satisfies the security needs of the vehicle network, and is resilient to general assaults (forgery assault, modify assault, MITM assault, replay assault, and quantum computing assaults). In conclusion, a thorough security analysis proved that VANETs' security and privacy standards would be met by the proposed scheme. Our more secure and efficient scheme was validated by a thorough analysis of the communicational and computational overhead it entails.

In future work, we carry out experiment results using simulation networks such as OMNeT ++ and traffic simulations such as SUMO. Meanwhile, this paper will extend to protect TA masquerading attacks.

## Author Contributions

**Formal analysis:** Abdulwahab Ali Almazroi, Mahmood A. Al-Shareeda.

**Funding acquisition:** Mohammed A. Alqarni.

**Investigation:** Abdulwahab Ali Almazroi, Mohammed A. Alqarni.

**Project administration:** Abdulwahab Ali Almazroi, Mohammed A. Alqarni, Mahmood A. Al-Shareeda.

**Resources:** Abdulwahab Ali Almazroi, Mohammed A. Alqarni, Selvakumar Manickam.

**Supervision:** Selvakumar Manickam.

**Validation:** Mohammed A. Alqarni, Selvakumar Manickam.

**Writing – original draft:** Mahmood A. Al-Shareeda.

**Writing – review & editing:** Mahmood A. Al-Shareeda.

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
