## [Decision Letter · Decision Letter 0]

27 Jun 2023

PONE-D-23-11761L-CPPA: Lattice-Based Conditional Privacy-Preserving Authentication Scheme for Fog Computing with 5G-Enabled Vehicular SystemPLOS ONE

Dear Dr. Al-Shareeda‬‏,

Thank you for submitting your manuscript to PLOS ONE. After careful consideration, we feel that it has merit but does not fully meet PLOS ONE’s publication criteria as it currently stands. Therefore, we invite you to submit a revised version of the manuscript that addresses the points raised during the review process.

We look forward to receiving your revised manuscript.

Kind regards,

Faouzi Jaidi

Academic Editor

PLOS ONE

Journal Requirements:

“The authors extend their appreciation to the Deputyship for Research \\&  Innovation, Ministry of Education in Saudi Arabia for funding this research work through project number MoE-IF-UJ-22-04100409-X”

Reviewers' comments:

Reviewer's Responses to Questions

**Comments to the Author**

1. Is the manuscript technically sound, and do the data support the conclusions?

Reviewer #1: Yes

Reviewer #2: No

Reviewer #3: Yes

2. Has the statistical analysis been performed appropriately and rigorously? 

Reviewer #1: Yes

Reviewer #2: No

Reviewer #3: N/A

3. Have the authors made all data underlying the findings in their manuscript fully available?

Reviewer #1: Yes

Reviewer #2: Yes

Reviewer #3: Yes

4. Is the manuscript presented in an intelligible fashion and written in standard English?

Reviewer #1: Yes

Reviewer #2: No

Reviewer #3: Yes

5. Review Comments to the Author

Reviewer #1: The work is interesting and timely. It's easy to follow the paper. There are a few improvements listed below that are suggested to improve the manuscript.

1. Performance in terms of percentage must be mentioned in the abstract.

2. To make the paper more readable and the results more convincing, the significance of the paper should be clearly explained relative to previous work. Besides, several of the critical background literature especially for fog computing and privacy preserving are missing including but not limited to "A provably secure key transfer protocol for the fog-enabled Social Internet of Vehicles based on a confidential computing environment" and "Privacy‐preserving authentication scheme for digital twin‐enabled autonomous vehicle environments".

3. The authors should provide an adversary model.

4. The conclusion section is not impressive. This section should include a very brief discussion on future works. What are the problems that go beyond the scope of this paper? What are the issues that need to be addressed in contemporary research?

5. A more thorough proofreading of this work would be beneficial.

Reviewer #2: The purpose of this paper is to devise a lattice-based conditional privacy-preserving authentication scheme for fog computing with 5G-enabled vehicular system. It has the following problems

- The proposed scheme is weak against TA masquerading attack by registered fog servers because TA preloads its master secret key E to all fog servers.

- It needs to clarify how a vehicle use ID and password.

- Cryptographic scheme design should be clearly defined every notations before they are used. However, there are lots of mistakes on the notations usage. Furthermore, functions used in this paper should be clearly defined.

- It is unclear that how Ci should be the same as D*SKi.

- The scheme does not provide freshness of message because it uses the same SKi for each session.

- There are a lot of mistakes on the notation usages.

- Thereby, it is not easy to follow and understand the authentication scheme. Is it a cryptographic scheme?

Reviewer #3: 1. Please describe the experimental results of this study in the abstract.

2. The authors only compare with [48] (2019) and [49] (2020). The authors should compare with recent publications, e.g., [50] (2022) and other Lattice-based conditional privacy-preserving authentication schemes for VANET.

3. The authors miss the Figure, "as shown in Figure ??", in Line 145 (Page 5).

4. The authors should emphasize why their conditional privacy-preserving authentication scheme for fog computing with 5G-enabled VANET. Other schemes are also can be applied to fog computing with 5G-enabled VANET.

6. PLOS authors have the option to publish the peer review history of their article (what does this mean?). If published, this will include your full peer review and any attached files.

Reviewer #1: No

Reviewer #2: No

Reviewer #3: No

---

## [Author Response · Author response to Decision Letter 0]

3 Jul 2023

Reviewer 1: Concern #1:. Performance in terms of percentage must be mentioned in the abstract.

Reviewer 1: Response #1: Thanks for the valuable comments. We revise the manuscript by adding the result in terms of computation cost and communication costs at the end of the abstract as follows.

“Finally, the computation cost of the proposed L-CPPA scheme in terms of message signing, single verification and batch verification is 694.161 ms, 60.118 ms, and 1348.218 ms, respectively. Meanwhile, the communication cost is 7757 bytes.”

Reviewer 1: Concern #2:. To make the paper more readable and the results more convincing, the significance of the paper should be clearly explained relative to previous work. Besides, several of the critical background literature especially for fog computing and privacy-preserving are missing including but not limited to "A provably secure key transfer protocol for the fog-enabled Social Internet of Vehicles based on a confidential computing environment" and "Privacy‐preserving authentication scheme for digital twin‐enabled autonomous vehicle environments".

Reviewer 1: Response #2: Thanks for the valuable comments. We revise the manuscript by adding these two relevant articles for defining fog computing and privacy-preserving in the introduction section as follows.

“Information overflow due to redundant data may occur in autonomous vehicles because of the necessity for real-time sensing, calculation, and communication. In addition, the limited range of current communication technologies makes it challenging to anticipate traffic conditions outside of a vehicle's line of sight. Autonomous driving conditions have seen the development of digital twin systems to address these concerns [9]. Sensitive information about drivers is easily intercepted and altered since the data created by automobiles in motion with servers is carried on public networks. Furthermore, servers are put under extreme strain by the vast quantities of real-time data produced by these vehicles, devices, drivers, passengers, and other social links. As a result, Chen et al. [10] presented a secure authentication technique that makes use of private cloud infrastructures. In addition, they provided a more efficient key transfer phase to lessen the burden of computation on cloud servers.”

Reviewer 1: Concern #3. The authors should provide an adversary model.

Reviewer 1: Response #3: Thanks for the valuable comments. We revise the manuscript by providing an adversary model in the section of preliminaries as follows.

This subsection defines the scope and limitation of the adversary model used in the proposed L-CPPA scheme for fog computing with a 5G-enabled vehicular system. The proposed LC-CPPA scheme should be resisted the following adversary model.

- Modify Assaults: A robust policy against impersonation attacks is one that effectively stops attackers from posing as legitimate users.

- Man-In-The-Middle (MITM) Assaults: To resist MITM, a third party should be not able to capture the message sent between the legal senders and receivers.

- Forgery Assaults: To prevent a forgery attack, the protocol must be able to expose the attacker's effort to fabricate the sent traffic.

- Replay Assaults: To prevent traffic from being recovered from one session and used in another, the protocol should incorporate a nonce or timestamp into each message.

- Quantum Assaults: To resist quantum attacks, traditional cryptography should be not used in the proposed. 

Reviewer 1: Concern #4:. The conclusion section is not impressive. This section should include a very brief discussion on future works. What are the problems that go beyond the scope of this paper? What are the issues that need to be addressed in contemporary research?

Reviewer 1: Response #4: Thanks for the valuable comments. We revise the manuscript by adding future work in the conclusion section as follows.

“In future work, we carry out experiment results using simulation networks such as OMNeT++ and traffic simulations such as SUMO. Meanwhile, this paper will extend to protect TA masquerading attacks.”

Reviewer 1: Concern #5:. A more thorough proofreading of this work would be beneficial.

Reviewer 1: Response #5: Thanks for the valuable comments. The manuscript has been proofread by a professional English service.

Reviewer 2: Concern #1:. The proposed scheme is weak against TA masquerading attack by registered fog servers because TA preloads its master secret key E to all fog servers.

Reviewer 2: Response #1: Thanks for the valuable comments. We revise the manuscript by providing the adversary model applied for this proposal and adding the assumption in the system model.

*In a subsection of the Adversary Model,

This subsection defines the scope and limitation of the adversary model used in the proposed L-CPPA scheme for fog computing with a 5G-enabled vehicular system. The proposed LC-CPPA scheme should be resisted the following adversary model.

- Modify Assaults: A robust policy against impersonation attacks is one that effectively stops attackers from posing as legitimate users.

- Man-In-The-Middle (MITM) Assaults: To resist MITM, a third party should be not able to capture the message sent between the legal senders and receivers.

- Forgery Assaults: To prevent a forgery attack, the protocol must be able to expose the attacker's effort to fabricate the sent traffic.

- Replay Assaults: To prevent traffic from being recovered from one session and used in another, the protocol should incorporate a nonce or timestamp into each message.

- Quantum Assaults: To resist quantum attacks, traditional cryptography should be not used in the proposed. 

*Besides, in a subsection of the proposed system model, under TA, (In this paper, we assume the TA component is very strong and reliable from masquerading attacks.).

Reviewer 2: Concern #2:. It needs to clarify how a vehicle use ID and password.

Reviewer 2: Response #2: Thanks for the valuable comments. We revise the manuscript by clearly explaining how we use vehicle ID and password in the proposed L-CPPA scheme section as follows.

During the registration step, the driver inputs the vehicle TID and Password PWD to TA for receiving system parameters and verification code. With this step, the vehicle is able to join the vehicular system to complete the next step as follows. Vehicles hide true identity TID by hashing verification codes to obtain anonymous identity AID, which uses publicly to exchange the message among vehicles or nearby fog servers via 5G-BS. While during message signing, the vehicles hide TID by hashing the verification code and current timestamp together. TID and AID refer to true identity and anonymous identity, respectively, which describes in Table 1.

Reviewer 2: Concern #3:. Cryptographic scheme design should be clearly defined every notation before they are used. However, there are lots of mistakes on the notations usage. Furthermore, the functions used in this paper should be clearly defined.

Reviewer 2: Response #3: Thanks for the valuable comments. We revise the manuscript by double-checking the notation usage and clearly defining the notation used in Table 1 as follows.

Reviewer 2: Concern #4:. It is unclear that how Ci should be the same as D*SKi.

Reviewer 2: Response #4: Thanks for the valuable comments. For more detail about the algorithm of SampleD, please see paper entitled (Trapdoors for hard lattices and new cryptographic constructions)

Reviewer 2: Concern #5:. The scheme does not provide freshness of message because it uses the same SKi for each session.

Reviewer 2: Response #5: We would like to clarify the concern as follows. In the message signing step, the vehicle multiplies Ski hashing value that included the anonymous-ID and current timestamp to obtain the signature key Pki, which each session has different timestamps for providing freshness message. Meanwhile, the verifier also checks the timestamp before accepting the messages.

Signer side:

Verifier side:

Reviewer 2: Concern #6: There are a lot of mistakes on the notation usages.

Reviewer 2: Response #6: Thanks for the valuable comments. We revise the manuscript by double checking for all notation usages and proving definition of them into table 1 as follows.

Reviewer 2: Concern #7:. Thereby, it is not easy to follow and understand the authentication scheme. Is it a cryptographic scheme?

Reviewer 2: Response #7: Thanks for the valuable comments. We revise the manuscript by adding briefly presentation used figure to easy understand the authentication scheme as follows.

Reviewer 3: Concern #1: Please describe the experimental results of this study in the abstract.

Reviewer 3: Response #1: Thanks for the valuable comments. We revise the manuscript by adding the result in terms of computation cost and communication costs in end of the abstract as follows.

“Finally, the computation cost of the proposed L-CPPA scheme in terms of message signing, single verification and batch verification is 694.161 ms, 60.118 ms, and 1348.218 ms, respectively. Meanwhile, the communication cost is 7757 bytes.”

Reviewer 3: Concern #2: The authors only compare with [48] (2019) and [49] (2020). The authors should compare with recent publications, e.g., [50] (2022) and other Lattice-based conditional privacy-preserving authentication schemes for VANET.

Reviewer 3: Response #2: Thanks for the valuable comments. This paper scopes and limitations of the research problem under only these schemes of [48] (2019) and [49] (2020) as mentioned as follows.

“The proposed L-CPPA scheme employs a fog server to generate and send a secret key for each registered vehicle via 5G-BS, which addresses the limitation arising in [52]. The TA in the proposed L-CPPA scheme saves its master private key to each fog server instead of the OBU of the vehicle, which addresses the limitation arising in [50, 51].”

Reviewer 3: Concern #3: The authors miss the Figure, "as shown in Figure ??", in Line 145 (Page 5).

Reviewer 3: Response #3: Thanks for the valuable comments. We revise the manuscript by adding figure to explain the proposed model as follows.

Reviewer 3: Concern #4: The authors should emphasize why their conditional privacy-preserving authentication scheme for fog computing with 5G-enabled VANET. Other schemes are also can be applied to fog computing with 5G-enabled VANET.

Reviewer 3: Response #4: Thanks for the valuable comments. We revise the manuscript by adding the research on how the proposed scheme outperforms other schemes in the related work section as follows.

---

## [Decision Letter · Decision Letter 1]

2 Aug 2023

PONE-D-23-11761R1L-CPPA: Lattice-Based Conditional Privacy-Preserving Authentication Scheme for Fog Computing with 5G-Enabled Vehicular SystemPLOS ONE

Dear Dr. Al-Shareeda‬‏,

Thank you for submitting your manuscript to PLOS ONE. After careful consideration, we feel that it has merit but does not fully meet PLOS ONE’s publication criteria as it currently stands. Therefore, we invite you to submit a revised version of the manuscript that addresses the points raised during the review process. Please submit your revised manuscript by Sep 16 2023 11:59PM. If you will need more time than this to complete your revisions, please reply to this message or contact the journal office at plosone@plos.org. Please include the following items when submitting your revised manuscript:A rebuttal letter that responds to each point raised by the academic editor and reviewer(s). You should upload this letter as a separate file labeled 'Response to Reviewers'.A marked-up copy of your manuscript that highlights changes made to the original version. You should upload this as a separate file labeled 'Revised Manuscript with Track Changes'.An unmarked version of your revised paper without tracked changes. You should upload this as a separate file labeled 'Manuscript'.If applicable, we recommend that you deposit your laboratory protocols in protocols.io to enhance the reproducibility of your results. Protocols.io assigns your protocol its own identifier (DOI) so that it can be cited independently in the future. For instructions see: https://journals.plos.org/plosone/s/submission-guidelines#loc-laboratory-protocols. Additionally, PLOS ONE offers an option for publishing peer-reviewed Lab Protocol articles, which describe protocols hosted on protocols.io. Read more information on sharing protocols at https://plos.org/protocols?utm_medium=editorial-email&utm_source=authorletters&utm_campaign=protocols.

We look forward to receiving your revised manuscript.

Kind regards,

Faouzi Jaidi

Academic Editor

PLOS ONE

Journal Requirements:

**Additional Editor Comments:**

The authors are asked to carefully address the reviewer’s comments as well as the following comments:

The presentation and readability of the paper should be enhanced:

- Ensure that your manuscript meets PLOS ONE's style requirements, including those for file naming. The PLOS ONE style templates can be found at

- Add numbers to all sections and subsections (with respect to the journal template).

- Correct missed section’s references /numbers in the last paragraph of the introduction “Our remaining tasks are divided into several parts: Section proposes the 60 classification of the related works. Section provides the background of the vehicular 61 network. …” .

- Check if all figures and tables are referenced and explained in the text.

- Check if all the references respect the structure required by the journal and are alphabetically ordered.

- A more thorough proofreading of this manuscript is needed “as example: analyse/analyze (British / US English)”.

Reviewers' comments:

Reviewer's Responses to Questions

**Comments to the Author**

1. If the authors have adequately addressed your comments raised in a previous round of review and you feel that this manuscript is now acceptable for publication, you may indicate that here to bypass the “Comments to the Author” section, enter your conflict of interest statement in the “Confidential to Editor” section, and submit your "Accept" recommendation.

Reviewer #1: All comments have been addressed

Reviewer #2: All comments have been addressed

2. Is the manuscript technically sound, and do the data support the conclusions?

Reviewer #1: Yes

Reviewer #2: Yes

3. Has the statistical analysis been performed appropriately and rigorously? 

Reviewer #1: Yes

Reviewer #2: Yes

4. Have the authors made all data underlying the findings in their manuscript fully available?

Reviewer #1: Yes

Reviewer #2: Yes

5. Is the manuscript presented in an intelligible fashion and written in standard English?

Reviewer #1: Yes

Reviewer #2: Yes

6. Review Comments to the Author

Reviewer #1: No further comments now. The editorial quality has been improved and I believe the paper's contribution warrants acceptance.

Reviewer #2: The authors corrected well for the first round of review. However, it still has the following simple problems to be sorted out.

- The abbreviation for the proposed scheme is L-CPPA. However, LC-CPPA is appeared in page 8.

- h3 is the third hash function, which should use subscript for 3 at pages 9 and 10.

- PWD and TID are defined in Table 1. However, contents use PWD_i and TID_i instead of them.

- T_1 is use for the timestamp. However, T_i is used for the definition of T_1 in page 11.

- Does the equation 6 require T_1 as a parameter for h_2 as the same as the equation 10?

7. PLOS authors have the option to publish the peer review history of their article (what does this mean?). If published, this will include your full peer review and any attached files.

Reviewer #1: No

Reviewer #2: No

---

## [Author Response · Author response to Decision Letter 1]

2 Aug 2023

Reviewer 2: Concern #1: The abbreviation for the proposed scheme is L-CPPA. However, LC-CPPA is appeared in page 8.

Reviewer 2: Response #1: Sorry for this mistake. Now the paper has revised this issue as follows. 

(The following is an adversary model that the proposed L-CPPA technique should be able to counter.)

Reviewer 2: Concern #2: h3 is the third hash function, which should use subscript for 3 at pages 9 and 10.

Reviewer 2: Response #2: Thanks, we revised in page 9 as well page 10 by using subscript for 3.

Reviewer 2: Concern #3: PWD and TID are defined in Table 1. However, contents use PWD_i and TID_i instead of them.

Reviewer 2: Response #3: Thanks, we revise the table by adding same on the content of the paper.

Reviewer 2: Concern #4: T_1 is use for the timestamp. However, T_i is used for the definition of T_1 in page 11.

Reviewer 2: Response #4: we have define of ti in table to be I=1,2,3,n

Reviewer 2: Concern #5: Does the equation 6 require T_1 as a parameter for h_2 as the same as the equation 10?

Reviewer 2: Response #5: Sure and thanks, T_1 has been added now in the paper.

---

## [Decision Letter · Decision Letter 2]

15 Aug 2023

PONE-D-23-11761R2L-CPPA: Lattice-Based Conditional Privacy-Preserving Authentication Scheme for Fog Computing with 5G-Enabled Vehicular SystemPLOS ONE

Dear Dr. Al-Shareeda‬‏,

Thank you for submitting your manuscript to PLOS ONE. After careful consideration, we feel that it has merit but does not fully meet PLOS ONE’s publication criteria as it currently stands. Therefore, we invite you to submit a revised version of the manuscript that addresses the points raised during the review process.

Based on the advice received, your manuscript could be accepted for publication, with minor adjustments. Please see the notes bellow. These are minor revisions, which I believe you can do very quickly but should be handled carefully.

The presentation and readability of the paper should be enhanced:

Ensure that your manuscript meets PLOS ONE's style requirements, including those for file naming. The PLOS ONE style templates can be found at https://journals.plos.org/plosone/s/file?id=wjVg/PLOSOne_formatting_sample_main_body.pdf and https://journals.plos.org/plosone/s/file?id=ba62/PLOSOne_formatting_sample_title_authors_affiliations.pdfAdd numbers to all sections and subsections (with respect to the journal template).Correct missed section’s references /numbers in the last paragraph of the introduction “Our remaining tasks are divided into several parts: Section proposes the 60 classification of the related works. Section provides the background of the vehicular 61 network. …” .Check if all figures and tables are referenced and explained in the text.Check if all references respect the structure required by the journal and are alphabetically ordered.A more thorough proofreading of this manuscript is needed and the writing of the paper should be reviewed carefully. The followings are examples of typos to correct  :At lines 13; 30 : " computing-based fog " do you mean : fog-based computing ?Line 70 : Several researchers [16–26] have been proposed PKI systems –-> Several researchers [16–26] have proposed PKI systemsLine 107 : sentence incomplete “Additionally, several researchers [41–47].” ; Al-Shareeda et al. [45] uses ECC - Al-Shareeda et al. [45] use ECC ;Line 139 : "Based on their knowledge," their it refers to whom? Do you mean "Based on our knowledge,"Line 157 : “are the the 5G-Base” - “are the 5G-Base”Line 251: “the true identify” Do you mean “the true identity”Line 275: “should be not used” - “should not be used”Line 394: “If Eq. 12 oks” - “If equation 12 is verified”Line 399: “information i, Msgi ∈ [n].” do you mean “information Msgi, i ∈ [n].”?Line 431: “TIDi is the identify of the vehicle” - “TIDi is the identity of the vehicle”Line 443: “Once obtainin” - “Once obtained”“analyse/analyze (British / US English)”.…Abbreviations should be explained during their first use: (line 78) TA is not previously explained.The presentation of equation 11 should be reviewed and enhanced. Please submit your revised manuscript by Sep 29 2023 11:59PM. If you will need more time than this to complete your revisions, please reply to this message or contact the journal office at plosone@plos.org. Please include the following items when submitting your revised manuscript:A rebuttal letter that responds to each point raised by the academic editor and reviewer(s). You should upload this letter as a separate file labeled 'Response to Reviewers'.A marked-up copy of your manuscript that highlights changes made to the original version. You should upload this as a separate file labeled 'Revised Manuscript with Track Changes'.An unmarked version of your revised paper without tracked changes. You should upload this as a separate file labeled 'Manuscript'.If applicable, we recommend that you deposit your laboratory protocols in protocols.io to enhance the reproducibility of your results. Protocols.io assigns your protocol its own identifier (DOI) so that it can be cited independently in the future. For instructions see: https://journals.plos.org/plosone/s/submission-guidelines#loc-laboratory-protocols. Additionally, PLOS ONE offers an option for publishing peer-reviewed Lab Protocol articles, which describe protocols hosted on protocols.io. Read more information on sharing protocols at https://plos.org/protocols?utm_medium=editorial-email&utm_source=authorletters&utm_campaign=protocols.

We look forward to receiving your revised manuscript.

Kind regards,

Faouzi Jaidi

Academic Editor

PLOS ONE

Journal Requirements:

Reviewers' comments:

Reviewer's Responses to Questions

**Comments to the Author**

1. If the authors have adequately addressed your comments raised in a previous round of review and you feel that this manuscript is now acceptable for publication, you may indicate that here to bypass the “Comments to the Author” section, enter your conflict of interest statement in the “Confidential to Editor” section, and submit your "Accept" recommendation.

Reviewer #2: All comments have been addressed

2. Is the manuscript technically sound, and do the data support the conclusions?

Reviewer #2: Yes

3. Has the statistical analysis been performed appropriately and rigorously? 

Reviewer #2: Yes

4. Have the authors made all data underlying the findings in their manuscript fully available?

Reviewer #2: Yes

5. Is the manuscript presented in an intelligible fashion and written in standard English?

Reviewer #2: Yes

6. Review Comments to the Author

Reviewer #2: Authors addressed the previous review comments well enough. I believe that this final manuscript is properly revised.

7. PLOS authors have the option to publish the peer review history of their article (what does this mean?). If published, this will include your full peer review and any attached files.

Reviewer #2: No

---

## [Author Response · Author response to Decision Letter 2]

8 Sep 2023

Reviewer#1, Concern # 1: Ensure that your manuscript meets PLOS ONE's style requirements, including those for file naming. The PLOS ONE style templates can be found at https://journals.plos.org/plosone/s/file?id=wjVg/PLOSOne_formatting_sample_main_body.pdf and https://journals.plos.org/plosone/s/file?id=ba62/PLOSOne_formatting_sample_title_authors_affiliations.pdf

Author response: Thanks for valuable comments. 

Author action: We updated the manuscript by rewrite the paper by using LATEX format for POLS ONE journal.

Reviewer#1, Concern # 2: Add numbers to all sections and subsections (with respect to the journal template).

Author response: Thanks for valuable comments. 

Author action: We updated the manuscript by adding numbers to all sections and subsections.

Reviewer#1, Concern # 3: Correct missed section’s references /numbers in the last paragraph of the introduction “Our remaining tasks are divided into several parts: Section proposes the 60 classification of the related works. Section provides the background of the vehicular 61 network. …” .

Author response: Thanks for valuable comments. 

Author action: We updated the manuscript 

Reviewer#1, Concern # 4: Check if all figures and tables are referenced and explained in the text.

Author response: Thanks for valuable comments. 

Author action: We updated the manuscript by checking

Reviewer#1, Concern # 5: Check if all references respect the structure required by the journal and are alphabetically ordered.

Author response: Thanks for valuable comments. 

Author action: We updated the manuscript by checking

Reviewer#1, Concern # 6: A more thorough proofreading of this manuscript is needed and the writing of the paper should be reviewed carefully. The followings are examples of typos to correct: At lines 13; 30 : " computing-based fog " do you mean : fog-based computing ? Line 70 : Several researchers [16–26] have been proposed PKI systems –-> Several researchers [16–26] have proposed PKI systems..Line 107 : sentence incomplete “Additionally, several researchers [41–47].” ; Al-Shareeda et al. [45] uses ECC - Al-Shareeda et al. [45] use ECC ; Line 139 : "Based on their knowledge," their it refers to whom? Do you mean "Based on our knowledge," Line 157 : “are the the 5G-Base” - “are the 5G-Base” Line 251: “the true identify” Do you mean “the true identity” Line 275: “should be not used” - “should not be used” Line 394: “If Eq. 12 oks” - “If equation 12 is verified” Line 399: “information i, Msgi ∈ [n].” do you mean “information Msgi, i ∈ [n].”? Line 431: “TIDi is the identify of the vehicle” - “TIDi is the identity of the vehicle” Line 443: “Once obtainin” - “Once obtained” “analyse/analyze (British / US English)”.

Author response: Thanks for valuable comments. 

Author action: We updated the manuscript by revising all above concern on the manuscript

Reviewer#1, Concern # 7: Abbreviations should be explained during their first use: (line 78) TA is not previously explained.

Author response: Thanks for valuable comments. 

Author action: We updated the manuscript by adding previously explained during their first use.

Reviewer#1, Concern # 8: The presentation of equation 11 should be reviewed and enhanced.

Author response: Thanks for valuable comments. 

Author action: We updated the manuscript by enhancing the presentation of equation 11.

---

## [Editor Report · Decision Letter 3]

27 Sep 2023

L-CPPA: Lattice-Based Conditional Privacy-Preserving Authentication Scheme for Fog Computing with 5G-Enabled Vehicular System

PONE-D-23-11761R3

Dear Dr. Al-Shareeda‬‏,

We’re pleased to inform you that your manuscript has been judged scientifically suitable for publication and will be formally accepted for publication once it meets all outstanding technical requirements.

Kind regards,

Faouzi Jaidi

Academic Editor

PLOS ONE
---

## [Editor Report · Acceptance letter]

17 Oct 2023

PONE-D-23-11761R3 

L-CPPA: Lattice-Based Conditional Privacy-Preserving Authentication Scheme for Fog Computing with 5G-Enabled Vehicular System 

Dear Dr. Al-Shareeda‬‏:

I'm pleased to inform you that your manuscript has been deemed suitable for publication in PLOS ONE. Congratulations! Your manuscript is now with our production department. 

Kind regards, 

on behalf of

Dr. Faouzi Jaidi 

Academic Editor

PLOS ONE